# Insights into the Chemical Biology of Childhood Embryonal Solid Tumors by NMR-Based Metabolomics

**DOI:** 10.3390/biom9120843

**Published:** 2019-12-08

**Authors:** Melissa Quintero Escobar, Mariana Maschietto, Ana C. V. Krepischi, Natasa Avramovic, Ljubica Tasic

**Affiliations:** 1Biological Chemistry Group, Department of Organic Chemistry, Institute of Chemistry, University of Campinas (UNICAMP), Campinas 13083-970, Brazil; meliquies@gmail.com; 2Laboratory of Blood Coagulation, Department of Medical Physiopathology, Hemocentro, University of Campinas (UNICAMP), Campinas 13083-878, Brazil; 3Research Center, Boldrini Children’s Hospital, Campinas 13083-884, Brazil; marianamasc@gmail.com; 4Department of Genetics and Evolutionary Biology, Institute of Biosciences, University of Sao Paulo (USP), Sao Paulo 05508-0970, Brazil; ana.krepischi@ib.usp.br; 5Institute of Medical Chemistry, Faculty of Medicine, University of Belgrade, Belgrade 11000, Serbia; natasa.avramovic@med.bg.ac.rs

**Keywords:** embryonal solid tumors, neuroblastoma, Wilms tumor, retinoblastoma, hepatoblastoma, central nervous system tumors, metabolomics

## Abstract

Most childhood cancers occur as isolated cases and show very different biological behavior when compared with cancers in adults. There are some solid tumors that occur almost exclusively in children among which stand out the embryonal solid tumors. These cancers main types are neuroblastoma, nephroblastoma (Wilms tumors), retinoblastoma and hepatoblastomas and tumors of the central nervous system (CNS). Embryonal solid tumors represent a heterogeneous group of cancers supposedly derived from undifferentiated cells, with histological features that resemble tissues of origin during embryogenesis. This key observation suggests that tumorigenesis might begin during early fetal or child life due to the errors in growth or pathways differentiation. There are not many literature data on genomic, transcriptomic, epigenetic, proteomic, or metabolomic differences in these types of cancers when compared to the omics- used in adult cancer research. Still, metabolomics by nuclear magnetic resonance (NMR) in childhood embryonal solid tumors research can contribute greatly to understand better metabolic pathways alterations and biology of the embryonal solid tumors and potential to be used in clinical applications. Different types of samples, such as tissues, cells, biofluids, mostly blood plasma and serum, can be analyzed by NMR to detect and identify cancer metabolic signatures and validated biomarkers using enlarged group of samples. The literature search for biomarkers points to around 20–30 compounds that could be associated with pediatric cancer as well as metastasis.

## 1. Introduction

Magnetic resonance spectroscopy is a nondestructive and quantitative technique, which is applied in molecular profiling to identify and quantify metabolites since 1970, but it was not until the late 1990s that the terms metabolomics and metabonomics were introduced in science [1]. Metabolomics is the measurement of the amount (concentration) of the complete set of metabolites with molecular masses lower than 2500 Da, such as amino acids and peptides (small oligopeptides), organic acids, then, lipids (fatty acids and derivatives), vitamins, among others. Metabolic fingerprints of an organism, fluid, tissue, or cell are a sum of overall effects, such as pathological conditions, genetic variations, and external stimuli (Figure 1) [2,3]. On the other hand, metabonomics is just a part of the metabolomics, and a quantitative measurement of differences in metabolites observed from the comparison among samples from at least two groups (one is control group) [4]. From now on we will refer to the two methods as metabolomics. 

Nowadays, the use of improved and modern techniques allow identification of many metabolites in great number of samples in a rapid and accurate way, producing data that can complement the analysis of other types of high-throughput molecular data derived from genetics, epigenetics, and proteomics [5]. This combined strategy sustains the complex process of identifying biomarkers for responses to specific physiological, or pathological conditions, and/or nutritional interventions, as well as for prediction of biological outcomes [6]. Along with gas (GC) or liquid chromatography (LC) coupled to a mass spectrometry (MS) or tandem mass spectrometry (MS/MS), nuclear magnetic resonance (NMR) is one of the main methods used in metabolomics [7], (for differences among the spectroscopy methods used in metabolomic analysis see the review of Griffin and Shockcor) [8], and has a variety of applications in cancer research (Figure 1). Among all of the -omics approaches, metabolomics can enable a suitable investigation of specific metabolic alterations in biochemical pathways in cancer development and progression, adding important information to improve cancer diagnosis and treatment [8,9]. In this review, we summarize the current state of knowledge concerning the contribution of metabolomics by NMR to the research of embryonal solid tumors, pointing to robust clinical applications of this methodology.

In cancer, different studies (for a complete listing of metabolomics approaches see the review papers by Gowda et al., and Fuss and Chen) [10,11] have revealed the potential role of metabolomics in broadening the understanding of pathophysiological processes, such as characterization of primary and metastatic tumors, and identification of predictive biomarkers for therapeutic responses; monitoring real-time cancer biological activities, which might help diagnosis; evaluating the detailed metabolic impact of the anticancer agents (particularly metal complexes) on the cells to obtain information regarding drug mechanisms and triggered cellular adaptations, thus potentially providing information on cellular resistance or positive response; characterizing chemical composition of biofluids, cell, or tissue extract [3]. For example, in studies of intact tissues such as biopsy material or tumor cells, high-resolution magic angle spinning (HR-MAS) NMR can be used to obtain complete biochemical information with minimum sample preparation requirements. It is also possible to use the same sample for subsequent histopathological analysis, and the quality of the spectra is comparable to those of liquid state NMR. Thus, HR-MAS technique enables direct sample analysis, which avoids extraction procedures that could affect the metabolic composition [12]. 

Metabolomics is an approach somewhat neglected in pediatric oncology research when compared to research involving adult cancer [13,14,15,16]. However, there are some studies using different fluids and tissue/cell samples in pediatric tumors, analyzed by liquid, semi- or solid-state NMR spectroscopy (Table 1) as well as other studies that used intact tissues or cell cultures applying HR-MAS NMR. These approaches are essential to shorten the diagnosis time especially in cancers for which early detection and screening, although challenging, would significantly impact therapeutic decisions as well as prognosis.

## 2. Embryonal Solid Tumors

Around 40% of all pediatric cancers account to solid tumors (Table 2) [28]. Many of them are so-called embryonal tumors, a heterogeneous group of cancers that exhibit in common a developmental origin from undifferentiated cells, and present histological features that resemble the forming fetus tissues [29]. Their genesis is likely based on arrest and/or defects in cellular pathways involved in cell growth and differentiation during pre- and post-natal periods, which result in retention of either embryonal or fetal tissue characteristics [30]. Pre- and post-natal periods of growth are regulated by controlled cell division and apoptosis. Therefore, developing cells during these periods are particularly prone to tumorigenesis [31] if any of the pathways that control them fail. 

In the last decade, several new applications of tumor biomarkers have further enhanced diagnosis and helped to monitor specific malignancies in childhood [32], but advances in the research of embryonal solid tumors are required to provide an early and accurate diagnosis, possibly leading to new, more effective, and safer cancer therapies.

The most common types of embryonal solid tumors are neuroblastoma, some central nervous system tumors (medulloblastoma, ependymoma, and rhabdoid tumor), Wilms tumor, retinoblastoma, hepatoblastoma, clear cell sarcoma of the kidney, primitive neuroectodermic tumor (PNET)/Ewing sarcoma, and few other tumors [28]. There are a few studies based on metabolomics by NMR obtained from some of these cancer types, which are summarized in Table 3 and briefly commented in the following sections.

### 2.1. Neuroblastoma (NB)

Up to 8–10% of all childhood cancers account for neuroblastoma [28]. Neuroblastoma (NB) is an enigmatic tumor derived from cells of neural crest with peculiar characteristics and behavior. It can be considered a disease of the first decade of life where children aging less than four years are propense of present this type of tumor. The outcome is closely related to patient’s age at diagnosis, children <6 months with localized tumors have excellent survival rates (>90% of patients) with extremely low or no cytotoxic therapy needed to be used [34]. However, 30% of patients aging >18 months at diagnosis will develop metastatic disease despite aggressive multimodality therapy [35,36]. Most of NB occur as isolated non-syndromic cases; less than 2% of cases occur in patients with a positive family history, and these patients tend to be diagnosed earlier and may have more than one primary tumor [28]. To understand molecular basis and behavior of NB, research has been done on its genetic and biological characteristics. For updates in NB, see the reviews of Newman and Nuchtern, and Newman et al., [37,38]. Still, data on metabolic alterations in this cancer are scarce. It was demonstrated that NB cells (SH-SY5Y) when treated with cytotoxic agents [39,40] undergo similar metabolic changes and show altered and increased ratios between lipid to choline when drug sensitive. Lipids and phospholipids have many important and still some unknown and complex functions, which need to be understood better [39,40] and not all could be detected by NMR.

When investigated by NMR analyses [25], patients with NB presented anomalies in the choline metabolic patterns that could be linked to disfunction in electron transport in mitochondria. As reported in combined results from MS and ^1^H-NMR, phosphatidylcholines higher concentrations and higher activity of phospholipase were seen in the choline metabolic pathway of the neuronal cells [25]. 

In 2011 [26], the metabolic profiles obtained from healthy tissue from adrenal medulla were compared to the metabolic profiles of NB from biopsy tissues by HR-MAS. Tumor tissues showed higher concentrations of some metabolites, such as acetate, creatine, glycine, and altered ratio of glutamine to glutamate when compared to adrenal medulla. The healthy control tissues showed to be richer in adrenalin and citric acid. Acetate and lysine were higher in NB patients that were older than 12 months.

On the contrary, greater amounts of glutamate, glycine, serine, glutathione, ascorbic acid, and *myo*-inositol were found in the NB samples from younger children. It was possible to link the NB stages (I–IV) with detected metabolites, for example, I–II stages showed similar characteristics when compared glycine, aspartate and creatine concentrations or ratio between glutamine and glutamate. NB stage IV showed similar quantities of creatine and acetate. During an average follow-up of seven years, the patients with better recovery showed greater amounts of succinate, aspartate, and glutathione, and the patients with a poorer prognosis (Table 3) showed higher concentrations of taurine and acetate. NB were very similar to healthy neural tissue, just different because of high concentrations of *myo*-inositol and altered ratio between glutamate and glutamine, similar to other embryonal tumors. Unique metabolic profiles were also reported for the embryonal tumors that were neuroectodermal derived [41].

### 2.2. Nephroblastoma or Wilms Tumor (WT)

Wilms tumor arises from the primitive metanephrogenic blastema and represents from 6% to 8% of childhood malignancies [42]. It shows a triphasic morphology, composed from epithelium, blastemal, and mesenchyma. The last component is made from embryonal immature mesenchyme and some other structures like adipose tissue and skeletal muscle. The blastema is rich in undifferentiated small blue round cells and its epithelium is made from also immature structures, such as tubules and glomeruloid bodies [43,44]. This heterogeneity is also reflected in gene expression profiles [45] and genomic alterations [44] with impact in risk stratification and treatment response [46]. The studies in WT have tried to define the groups of patients of high and low risk, independently of the morphology and staging, using molecular markers [47,48,49]. The identified biomarkers have failed to give the consistent predictive information in regard to the clinical outcome of WT; however, data from NMR are scarce. WT is characterized by altered mitochondrial energy metabolism when different intratumor elements were compared. For example, blastema and epithelium showed normal mitochondrial mass, but stroma regions were depleted in mitochondria compared to normal kidney tissue, showed low porin expression and citrate synthase activity, as well as loss of all OXPHOS complexes [43]. There is also data on urine NMR metabolomics. Urine of ten healthy boys matched in gender, age, and race with WT cancer patients (91 urine samples from 4-stage WT cancer patients’ groups) were compared [50]. Urine samples from WT patients showed alterations and increase in amino acids (Ala, Leu, Ile), iso-valerate, 2-hidroxibutyrate, 2-oxoisovalerate, glucose, dimethylamine, and 2-oxoglutarate. On the other side, WT patients showed decrease in creatine, creatinine, acetate, and citrate in urine. 

Patients with Wilms tumor are characterized by high cure rate of around 85% (for recent advances in the management of WT see review of Lopes and Lorenzo) [51] and its treatment counts on surgery and chemo- and radio- therapies. But WT shows somewhat elevated rates of relapse, as great as 50% for diffuse anaplasia cases to 15% for young patients that presented favorable histology during first WT diagnostics and treatment. Understanding malignancy of WT has been improved because of the studies that explored genetics and molecular biology providing a set of tumor markers linked to secretion of hormones (erythropoietin and renin), such as hyaluronic acid and carcinoembryonic antigen. Finding altered metabolites in WT especially ones that differentiate therapy resistant blastemal cells would be of ultimate importance for prognostics purposes as to identify presence of the very aggressive blastema that do not respond to known chemotherapy.

### 2.3. Hepatoblastoma (HB)

Most of the hepatoblastoma patients are very young and aged below three years old [52]. This type of tumor is the third most common abdominal solid tumor, and only 10% of the cases can be linked to germline mutations [53,54]. Due to the rarity, genetic predisposing factors for HB are still unrevealed. Additionally, HB presents a relatively stable genome with just few genetic alterations reported recently [55,56]. Clinical trials examining multimodal therapy, which consists of neoadjuvant chemotherapy followed by complete surgical resection and subsequent cycles of chemotherapy, have led to improved outcome for children with HB [57]. Currently, the alpha-fetoprotein (AFP) level is the only biomarker used in the HB diagnosis, risk stratification, and monitoring of HB [58]. A markedly elevated AFP level suggests the presence of the disease, but concentrations of AFP may also be elevated in patients with hepatocellular carcinoma, germ cell tumors, as well as in benign liver tumors, including mesenchymal hamartoma and infantile hemangioma. Of note, elevation of AFP levels is normal in healthy infants and declines gradually until 8 months of age, rendering AFP more difficult to interpret in young children [35]. Therefore, metabolic biomarkers for HB or HB metabolic fingerprints would be an extraordinary add-in for diagnostics and cancer follow up.

### 2.4. Retinoblastoma (RB)

Retinoblastoma patients are also very young, with 75% of them not older than two years, which commonly show good outcomes if treatment of the malignancy starts on time [59]. When RB is found in both eyes, patients are even younger and usually around 12 months old. This bilateral RB corresponds to germline mutations in *RB1* gene and accounts for 25% of all RB cases. Also, children that carry germline *RB1* mutation have increased risks of developing other cancers, such as osteosarcoma, soft tissue sarcomas, or melanoma. Therefore, primary care providers should ensure that families and patients diagnosed as having retinoblastoma seek genetic counseling and molecular testing [28]. 

Kohe et al. [59] studied RB at metabolic levels and identified three principal metabolic subgroups within this tumor. The authors had also correlated RB histopathology and clinical features with metabolomics findings and linked cell differentiation, necrosis and invasion with taurine, lipids and phosphocholine, respectively [41,59]. Tumor location influences greatly its metabolic profile, for example, healthy retina is metabolically very active tissue as it has increased energy demand and thus high levels of metabolites involved in ATP production.

### 2.5. Central Nervous System (CNS) Tumors

The CNS tumors are found in the posterior fossa and within cerebellum. Pilocytic astrocytoma (PA), medulloblastoma, and ependymoma [1] are the commonest cerebellum tumors but some CNS tumor cases account for atypical tumors, such as teratoid/rhabdoid (ATRT) tumor [19]. CNS tumors need combined treatments, with safe surgery and chemo- and radio- therapies. It is needed to evaluate brain resection as to eliminate tumor tissue followed by radiotherapy and chemotherapy, but order could be inversed. Survival rates for CNS tumor patients over the years improved, but yet diagnostics biomarkers are unknown, and metabolomics can improve understanding of tumor pathways alteration and molecular basis and improve prognosis and tumor treatments follow-up. 

CNS and brain disorders often show imbalances in metabolites and neurotransmitters related to brain functions [41]. A great majority of NMR-based metabolomics studies on pediatric brain tumors had been conducted in vivo or ex vivo. Usually, the lists of altered metabolites in CNS tumors count on prognostic markers for patients with good survival rates: choline-containing compounds, *N*-acetyl alanine (NAA), phosphocreatine and creatine, lactate, lipids, glutamine to glutamate ratio (Gln/Glu), and glycine (Gly). Changes in the levels of the cited metabolites were also reported as sensitive parameters for differentiating pediatric brain tumors from normal tissues, and for disease progression monitoring [60,61]. Alanine (Ala), glutamate (Gln), aspartate (Asp), and metabolic pathways involved with these amino acids were cited as metabolites with the highest importance for the three most common CNS tumors [18]. *Myo*-inositol and NAA are commonly detected and identified as altered in brain tumor tissues, especially in tumors of glial origin [22]. Importance of the creatine, *myo*-inositol and NAA as neural metabolic markers is even more emphasized as these could be used for brain regions identifications [62]. Other neurotransmitters and amino acids, namely glutamine (Gln), glutamate (Glu), gama-aminobutyric acid (GABA) also influence tumor metabolism. Taurine (Tau) as well had been pointed to be elevated in embryonal tumors [41] and is one of the most cited biomarkers in pediatric cancer research. 

#### 2.5.1. Medulloblastoma (MB)

Medulloblastoma (MB) is dominant primary malignant brain and spinal tumor that begins with errors in progenitor cells differentiation (fetal cells) during brain forming. It shows good prognostics with 45% to 95% of cure, which correlate well with MB subgroups [63] based on differences in its molecular alterations. Cerebellum is rich in metabolites related to neural functions and that is why MB is likely to reflect neural origin [41]. Ex vivo metabolite profiling of MB tissue with HR-MAS had shown that MB can be discriminated from other childhood brain tumors by evaluating glutamate as a prognostic marker [22]. Furthermore, high taurine, phosphocholine and glycine distinguished MB too (Table 3) [19]. HR-MAS analysis showed that MB was characterized by high levels of cholesterol (Cho), glycerophosphocholines (GPC), taurine (Tau), a slightly increased level of *myo*-inositol, low levels of some fatty acids, and decreased level of NAA and Cr [23]. 

Medulloblastoma is comprised of four subgroups identified by molecular studies (transcriptomics and genomics) that correlate with clinical and prognostic parameters: Wnt signaling pathway activated, Sonic hedgehog pathway (SHH) activated, Group 3, and Group 4. Among the differences, most pediatric patients from Groups 3 and 4 have *MYCN* amplification and lower survival rates when compared to patients from subgroups Wnt and SHH, which usually have two copies of the gene [63]. These groups also present differences in the metabolic profile with *myo*-inositol, creatine, taurine, lipids, and aspartate used to discriminate Groups 3 and 4 from Groups Wnt and SHH [64]. 

#### 2.5.2. Astrocytoma (ACs)

They develop from astrocytes, which are known for their shapes in star format. ACs show different scales (I–IV), which depend on cells’ abnormal shape. For example, slowly growing and localized tumors are rated as low-grade ACs. Around 25% of the ACs are low grade tumors named pilocytic astrocytoma (PAs). PAs start as errors in development of cerebellum or optic pathways, where tumors occur as isolated and/or type I neurofibromatosis syndrome. Sometimes, ACs may develop in the thalamus, brainstem, and the spinal cord. It is rare to find PAs in patients older than 18 years [65]. On the other side, older patients show ACs rated as II–IV-grade, ones that grow very fast and need a severe treatments course.

The location of tumor, its subtype and grade, and stepwise progression from lower to higher grade, vary greatly with age. Magnetic resonance spectroscopy (MRS) [66] had been successfully applied in providing the prognostic information on high-grade astrocytoma by monitoring choline and lactate. Also, when glioma cell lines and rodent tumor models studied by MRS, biomarkers for specific treatments [66,67] have been found as very useful. Additionally, high concentrations of some fatty acids, then isoleucine (Ile), leucine (Leu), valine (Val), glutamate (Glu), *N*-acetyl alanine (NAA), and gama-aminobutyric acid (GABA), low concentrations of creatine (Cr), *myo*-inositol, and taurine (Tau) were found for this tumor [23] by high-resolution magic-angle spinning NMR. 

#### 2.5.3. Ependymoma (ED)

Ependymoma (ED) is a rare type of cancer in brain or spinal cord. It starts in cells that line the ventricles (fluid-filled spaces in the brain) as well as in the canal that holds the spinal cord. It is the third most common brain tumor in children. 90% of ED are intracranial and 65% of ED arise within the posterior fossa [68]. 50% of the ED are diagnosed underage of three years. Currently, stratification does not implicate in treatment changes, and the long-term prognosis remains poorly understood [69]. Key differences in the metabolite profiles for the main types of childhood cerebellar tumors (Table 3) differentiate the ED by high signals of *myo*-inositol and glycerophosphocholine levels [23,24], and made a successful classification for glial-cell (astrocytoma-ependymoma) versus non-glial-cell (medulloblastoma) tumors [24].

## 3. Metabolomics on Embryonal Solid Tumors

Regarding sample sizes used in metabolomics on embryonal solid tumors, the number of individuals included in the research and the age of investigated cases, differ. It is worth stating that high resolution magic angle spinning (HR-MAS) sample size can vary from 12 µL to 50 µL (depends on rotor that was used) and usually are around 15 mg [21] or 60 mg for the 4 mm rotors, liquid samples’ size varies from 250 to 500 µL, and cancer cell lines usually report preparation procedures starting from 10^6^ to 10^9^ cells per sample for 4 mm 50 µL rotors. Some articles explored the small number of individuals because of the rarity of embryonal solid tumors, such as 5–15 samples, while cohorts with a greater number of individuals counted on 90–250 samples [18,19,20,21,22,23,24,25,26,27,28,29,30,31,32,33,34,35,36,37].

The literature on solid tumors in children presents many altered metabolites in cancer, when analyzed and compared to healthy tissues. But most common alterations were reported for metabolites in lipid metabolism pathways, TCA cycle, branched-chain amino acids (BCAA) metabolism pathways, glycolysis, and oxidative stress responses. The main metabolic alterations in analyzed cancer types can be resumed as sketched in Figure 2.

For example, if taurine present in greater quantities in the cancer sample, it means that cysteine (Cys) and consequently methionine (Met) were used for its synthesis, thus, glutathione concentration would be lower and indicative for the poorer cell response to oxidative stress. More acetate would imply on higher Acetyl-CoA concentration and, higher amounts of fatty acids (FAs), phospholipids (PL) or cholesterol (Chol) may also be observed. Taurine and acetate were indicated as metabolites found in childhood cancer patients with poorer recovery prognostics.

## 4. Conclusions

Metabolomics by NMR is being applied with great success in many areas of research, including cancer research and management. This -omics technology explores tissues, cell samples, fluids, mostly blood plasma, and serum, to discover cancer signatures (Figure 3) that can be applied in diagnosis and the treatment follow ups. The method and consistency of collection and storage of samples are of paramount importance, as handling samples in systematically equal ways can overcome errors and avoid results with biased outcomes. It is also important to compare the cancer samples with at least one group that represents a different condition, in general the normal one, denominated healthy control group, and gender and age similarities among groups must be corrected, even in the case of pediatric patients [70]. 

Different pathological conditions can be also compared, such as primary and metastatic tumors, searching for dissimilarities or specific signatures. An ideal sampling should involve a group of tissues from the cancer zone and a group of healthy tissue from the same patient (biopsy or surgery), which can be compared with a group of the same tumor type with a metastatic behavior. Biofluid samples from the three groups to complete the first cohort are highly desirable. 

Pediatric solid tumors have diverse molecular and cellular features that reflect their unique cellular origins. With the recent completion of sequence analysis of more than 1,000 pediatric solid tumor genomes, a broad understanding of the genomic landscape of childhood cancers emerged, such as for cancers derived from mesodermal, ectodermal, and endodermal lineages [34]. Due to their molecular, cellular, developmental, and genetic diversity, pediatric solid tumors provide an ideal platform to study why these cells are more susceptible to malignant transformation at certain developmental stage than others. Literature search for biomarkers points to around 20–30 compounds that could be associated with pediatric cancer as well as metastasis, and the most cited ones are glucose, lactate, acetate, some amino acids such as glycine (Gly), serine (Ser), glutamine (Gln), glutamate (Glu), in addition to the ratio between the last two Glu/Gln, lipids, such as cholesterol and phospholipids, choline, *myo*-inositol, taurine (Figure 3). Still, the limited number of clinical trials on pediatric solid malignancies, combined with the difficulty in discriminating and diagnosing solid tumors by using standard techniques, have hindered progress in this area. Therefore, it is imperative not only to accurately determine the type of tumor but also detect tumor recurrences. NMR techniques offer great promise in the management of children with solid malignancies, providing insights in disease diagnosis and prediction of treatment response.

While a complete tumor resection is not always possible due to either the infiltrating nature or the anatomical location, or even a biopsy to confirm diagnosis can be challenging, to follow up the progression or response to treatment of these tumors is of ultimate importance for a proper management of the patients. An alternative is the development and use of the liquid biopsy based on metabolomics. Circulating biomarkers in the blood must be tumor specific and be present in detectable concentrations. Liquid biopsy encompasses a broad spectrum of approaches aimed at characterizing different components of body fluids, including circulating tumor cells, tumor cell-free DNA, circulating RNA, microRNAs, and extracellular vesicles [71]. For some tumors, metabolites are being used to monitor patients, similar to hepatoblastoma, where an elevated level of AFP is considered both a biomarker and prognostic factor [56]. However, for the vast majority of embryonal tumors, studies are still ongoing; the major aims are to disclose a profile or a unique metabolite with high sensitivity and sensibility to be used as diagnostic tool, to stratify patients according to their response to standard treatment, and to monitor tumor progression.

Other analytical methods are also used in research of childhood embryonal solid tumors, such as based on mass spectrometry and more information on MS-based research can be found elsewhere [72,73]. 

Finally, the main observations, according to the up-to-date NMR-metabolomics research results in pediatric cancer, could be resumed as follows. NMR-metabolomics provide an understanding of metabolic processes prevalent in a particular tumor type and how they are altered in cancer samples (body fluids or tissues) in relation to controls; but, still, we need to understand better how to correlate a set of metabolites identified as altered (up or down) in relation to cancer metabolic and molecular pathways. Detected set of biomarkers may be specific to a tumor subtype, or generic, such as an altered profile, regardless of the cancer type. If generic, biomarker need not even to be understood from the point of view of tumor biology, just to be specific for one tumor type, enabling detection and discrimination of tumor from healthy samples. Undoubtedly, the use of a liquid biopsy is vital as a non-invasive approach and extremely useful in cases in which biopsy is not possible. 

## Figures and Tables

**Figure 1 biomolecules-09-00843-f001:**
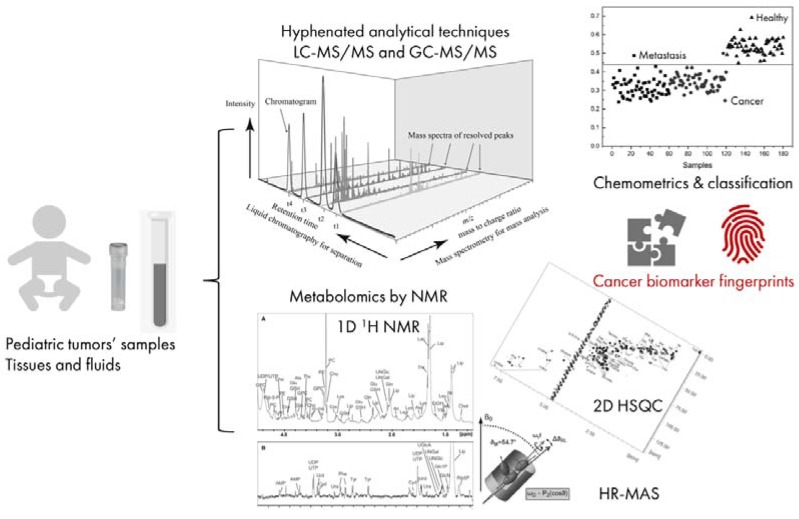
Graphical representation of metabolomics studies in childhood cancer research. Samples derived from patients can be tumor tissue or body fluids (at left). The analyses start with sample preparation for hyphenated analytical techniques, such as gas (GC) or liquid chromatography (LC), coupled with mass spectrometers (MS) as detectors as in tandem mass spectrometry (MS/MS); or with liquid or semisolid nuclear magnetic resonance (NMR) or high-resolution magic angle spinning (HR-MAS) NMR performed in one- and two-dimensional (1D and 2D). At upper right corner of the figure, an example of an ideal set of samples is depicted: a group of samples could be classified in three main classes, healthy/control, primary tumors, and metastasis, in accordance to the profile of their metabolites, which allows the characterization of a set of biomarkers that can be used as hallmarks of cancer or cancer fingerprints.

**Figure 2 biomolecules-09-00843-f002:**
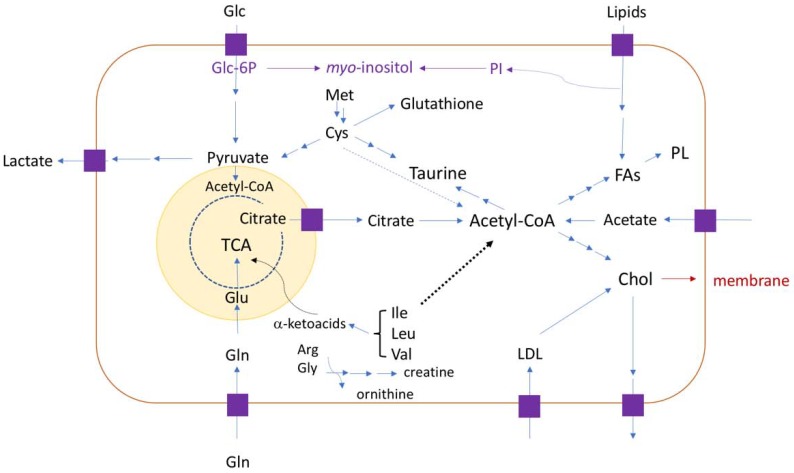
Illustration of the main metabolic pathways reported as altered in childhood solid tumors. Phospholipids (PL), cholesterol (Chol), Fatty Acids (FA), Phosphoinositol (PI), Acetyl-CoA, Branched-chain Amino Acids (BCAA, such as isoleucine (Ile), leucine (Leu), valine (Val)), Citrate, Taurine, Glycose (Glc), Lactate, Pyruvate, Glutamine (Gln), Glutamate (Glu), and creatine. For example, BCAA can provide alpha-ketoacids for TCA cycle or Acetyl-CoA. Cancer cells have high lactate concentration that is derived from pyruvate or cysteine (Cys), if Cys was used as precursor for lactate synthesis, then there would be decreases in the glutathione and taurine concentrations. Mitochondrion is shown in light yellow.

**Figure 3 biomolecules-09-00843-f003:**
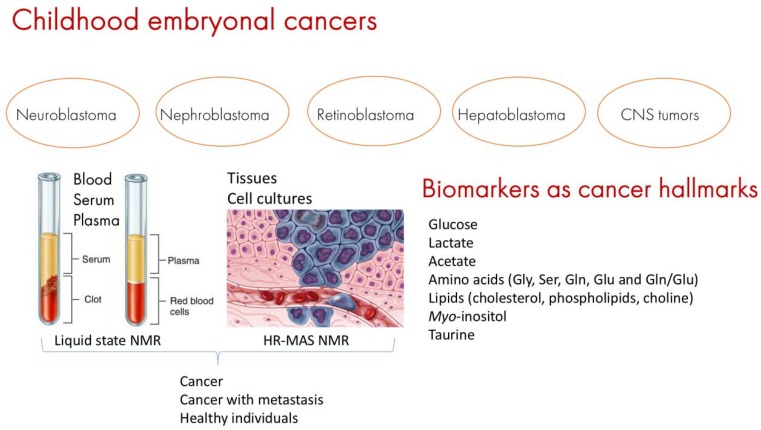
Discovering childhood embryonal cancer metabolic fingerprints by NMR. Ideally, samples should belong to three groups: (1) cancer samples, (2) cancer samples with metastatic behavior, and (3) healthy controls (for example tissue near the cancer zone which shows no histological alterations). Biofluids, tissues and cells must be handled in the same way—from sampling and storage to analyses. The most cited biomarkers are 20–30 compounds among which figure glucose, lactate, acetate, and some amino acids such as glycine, serine, glutamate, glutamine, leucine, isoleucine and valine, creatine and phosphocreatine, lipids (glycerophosphocholines, cholesterol, and some fatty acids), taurine and *myo*-inositol.

**Table 1 biomolecules-09-00843-t001:** Overview of some research studies in pediatric tumors using metabolomics.

Pediatric Tumor Type	Sample Type	Analytical Platforms	Research Area
Acute lymphoblastic leukemia (ALL)	Plasma	^1^H-NMR	Lipids [17]
Acute lymphoblastic leukemia (ALL)	Blood	HR-MAS	Tumor microenvironment [18]
Atypical teratoid /rhabdoid tumors	Tissue	HR-MAS	Metabolic profiles [19]
Brain tumor (astrocytomas and medulloblastoma)	Tissue	^1^H-NMR, HR-MAS	Metabolic characterization [20,21]
Brain and nervous system	Tissue	HR-MAS	Metabolic differences [22]
Cerebellar ependymoma	Tissue	HR-MAS	Metabolic profiles [19]
Ependymoma	Tissue	HR-MAS	Metabolic characterization [19,23]
Medulloblastoma	Tissue	HR-MAS, ^1^H-NMR	Metabolic characterization [19,23,24]
Neuroblastoma	Cell Cultures	^1^H-NMR	Anomalous choline metabolic patterns [25]
Neuroblastoma	Serum	^1^H-NMR	Utility of metabolomics in xenograft models
Neuroblastoma	Tissue	HR-MAS	Metabolomic profile [26]
Osteosarcoma	Cell Cultures	HR-MAS	Effect of cisplatin on the metabolic profile [27]
Osteosarcoma	Cell Cultures	^1^H-NMR	Metabonomics to monitor anticancer drugs [24,25]
Pilocytic astrocytoma	Tissue	HR-MAS, ^1^H-MRS	Metabolic characterization [18]

**Table 2 biomolecules-09-00843-t002:** Malignant solid tumors—principal types of embryonal solid tumors [33].

Localization	Pediatric Tumor Type	Age of Presentation (Year)
Central Nervous System	Medulloblastoma, astrocytoma, ependymoma Atypical teratoid/rhabdoid tumors (ATRT)	0–25 0–2
Liver	Hepatoblastoma	0–2
Kidney	Nephroblastoma or Wilms tumor Rhabdoid tumor	2–3 0–2
Sympathetic Nervous System	Neuroblastoma	0–4
Bone	Osteosarcoma Ewing Sarcoma	10–18
Soft Tissue	Rhabdomyosarcoma	2–8
Eye	Retinoblastoma	0–2

**Table 3 biomolecules-09-00843-t003:** Overview of the main metabolic findings in studies of childhood embryonal solid tumors by NMR.

Tumor Type	Sample	Metabolic Changes	Observations	Ref
**Neuroblastoma**	Tissue *	↑acetate, lysine	>12 months	[21]
		↑glycine, glutamine, glutamate, *myo*-inositol, serine, citric acid	<12 months	[21]
		glutamine/glutamate ratio, ↑aspartate, creatine, glycine, *myo*-inositol	Stages I–II	[21]
		↑acetate and creatine	Stage IV	[21]
		↑acetate and taurine	Poor prognosis	[21]
		↑aspartate, succinate, glutathione ↑lipids, NAA, *myo*-inositol, aspartate ↑taurine	Better prognosis Poor prognosis Poor prognosis	[21] [36] [21]
	Cell lines **	phosphatidylcholine, choline, glutamate, glutamine and branched chain amino acids ↑lipids/choline ratio and phospholipids ↑↓lipids	Mitochondria dysfunction Drug sensitive cells, when treated with cytotoxic agents Growth factor modulations	[20] [34] [35]
**Hepatoblastoma**	HepG2 cells	↑acyl groups of fatty acids, cholesterol, lactate, glycine, choline, phosphocholine, glycerophosphocholine (GPC), betaine, trimethylamine *N*-oxide (TMAO), hydroxyproline, branched-chain amino acids (BCAA), and glutamate ↓formiate ↑amino acids and energy metabolites ↓glutathione	Aflatoxin M1 Effects Metabolic modulations by Bisphenol A and 17β-Estradiol	[37] [38]
**Central nervous system** **Medulloblastoma**	Tissue *	↑glutamate ↑citrate, aspartate, ↑phosphocholine, taurine glucose, *scyllo*-inositol ↑phosphocholine, glycine, creatine ↓glutamine ↓NAA	Prognostic brain marker Developmental stages Possible markers of malignancy Developmental stages	[17] [18] [36] [23] [18]
**Astrocytoma**	Tissue *	↑glutamine ↑NAA, fatty acids, amino acids (Ile, Leu and Val), GABA, Glu ↓Cr, *myo*-inositol, taurine	Tumor identification Prognostic biomarker	[23] [18]
**Ependymoma**	Tissue *	↑*myo*-inositol, glutathione ↓leucine, glutamine ↓taurine	Developmental stages Prognostic biomarker	[18] [19]
**Retinoblastoma**	Tissue *	↑taurine ↑lipids ↓phosphocholine ↑GABA, Creatine ↓*myo*-inositol ↓glycine, NAA	Differentiation Necrosis Invasion Energy Metabolism Cell signaling Poor survival	[36] [36] [36] [37] [37] [37] [37]

* Tissue samples were around 15 mg for 4 mm rotors of 12 μL up to 60 mg for 4 mm rotors of 50 μL. ** Usually 10^6^–10^9^ cells were used for comparative studies (for example, 5 × 10^5^ cell pellets (40 μL) for 4 mm 50 μL rotors).

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
