# Peer review of "Insights into the Chemical Biology of Childhood Embryonal Solid Tumors by NMR-Based Metabolomics"

_biomolecules, 2019, doi:10.3390/biom9120843_

Round 1

Reviewer 1 Report

The paper by Escobar et al. is an interesting and comprehensive review of the available metabolomic literature in the field of embryonal solid tumours in the childhood. The authors limited the survey to the contributions based on NMR metabolomics, both in biofluids and in tissues.

This field is not particularly addressed in the literature, with scarce contributions with respect to, i.e., solid tumours in adults. Thus, this review could also contribute to stimulate further researches in this important area.

The only comment I have concerns the layout of table 3, which, being an outlook of the main findings retrieved from the literature, is the core of the review. For each paper listed in the table, I suggest to add also information regarding: the number of samples analysed in each compared group, the statistical analysis strategy employed (very briefly, just for quickly catching the experimental design) and the average age of the participants, to understand whether the data were collected on infants or toddlers. Further, I noticed that some papers cited in the main text are not included in the table. I understood that the table lists only the “main findings”, however due to the not so large corpus of papers, I encourage the authors to make the table more complete.

Author Response

Thank you for the suggestions. All recommendations were followed, and the table was amended as required. The added new information consists of the number of samples, age of the patients and chemometrics methodology cited in the listed publications. We agree that the added data contribute greatly to the manuscript's understanding.

Reviewer 2 Report

In the present review titled " Insights into the chemical biology of childhood enbryonal solid tumors by NMR-Based metabolomics" the authors have reviewed the literature pertaining to metabolites identified in solid tumors of embryological origins by NMR based metabolomics methods.

Overall impression: The review seems to be properly written in appropriate sections, the information on the metabolites related to tumors is well tabulated for easy understanding. However, there seems to be few minor areas where little work is needed before considering a publication,

1) Authors seems to have focused  mainly on metabolites identified using NMR approaches, however it would be more informative for readers if authors had also included or tabulated other metabolites identified through other analytical methods (LC/GC-MS) inconjunction with NMR based metabolomics, to cover more comprehensive data on identified metabolites/biomarkers. If description of such studies is beyond the scope of this paper, atleast references to those studies should be included.

2) Including a figure depicting role of metabolic pathways, related to common metabolites/biomarkers, in embryonal cancer development/progression would be more informative.

3) There are few typos in the paper which needs to be corrected.

lines 145 and 146,

table 3: phosphotidylcoline to phosphotidylcholine, ependymona to ependymoma.

4) There seems to be few reports of many urinary metabolites identified using NMR for nephroblastoma which are not covered in the review and need to be included.

MacLellan, Dawn, et al. "Urinary metabolite profiling by nuclear magnetic resonance spectroscopy to distinguish control patients from Wilms tumor (WT) and WT tumor by stage." (2014): e21013-e21013.

Author Response

We are grateful for the recommendations, suggestions, and critics of our manuscript. First, we have tried to correct the spelling and grammar mistakes and accept apologies for the errors made, as none of the authors in a native English speaker, therefore, some errors pass unnoticed.

Our manuscript was based strictly on NMR, as our group uses just from time-to-time MS, but we plan to write up a review on MS/MS data on the same cancer types and compare the data from multiple platforms. Some instructions on important articles in this field (MS-based metabolomics) were cited in the edited manuscript version.

We have included the Figure with altered pathways in cancer of the youngest patients reported so far in the literature.

Also, the recommended article was included in the list.

We hope that this manuscript version suits better the Biomolecules” standards.